# $\chi$-Model: Improving Data Efficiency in Deep Learning with a Minimax Model

**Ximei Wang, Xinyang Chen, Jianmin Wang, Mingsheng Long (✉)** *

School of Software, BNRist, Tsinghua University, China

wxm17@mails.tsinghua.edu.cn, chenxinyang95@gmail.com
jimwang@tsinghua.edu.cn, mingsheng@tsinghua.edu.cn,

## Abstract

To mitigate the burden of data labeling, we aim at improving data efficiency for both classification and regression setups in deep learning. However, the current focus is on classification problems while rare attention has been paid to deep regression, which usually requires more human effort to labeling. Further, due to the intrinsic difference between categorical and continuous label space, the common intuitions for classification, *e.g.* cluster assumptions or pseudo labeling strategies, cannot be naturally adapted into deep regression. To this end, we first delved into the existing data-efficient methods in deep learning and found that they either encourage invariance to *data stochasticity* (*e.g.*, consistency regularization under different augmentations) or *model stochasticity* (*e.g.*, difference penalty for predictions of models with different dropout). To take the power of both worlds, we propose a novel $\chi$-Model by simultaneously encouraging the invariance to data stochasticity and model stochasticity. Extensive experiments verify the superiority of the $\chi$-Model among various tasks, from a single-value prediction task of age estimation to a dense-value prediction task of keypoint localization, a 2D synthetic and a 3D realistic dataset, as well as a multi-category object recognition task.

## 1 Introduction

In the last decade, deep learning has become the *de facto* choice for numerous machine learning applications in the presence of large-scale labeled datasets. However, collecting sufficient labeled data through manual labeling, especially for deep regression tasks such as keypoint localization and age estimation, is prohibitively time-costly and labor-expensive in real-world scenarios. To mitigate the requirement of labeled data, great effort (Lee, 2013; Laine & Aila, 2017; Grandvalet & Bengio, 2005; Sohn et al., 2020; Chen et al., 2020b) has been made to improve data efficiency in deep learning from the perspectives of simultaneously exploring both labeled and unlabeled data based on the intuitions for classification, *e.g.* cluster assumptions, pseudo labeling strategies, or consistency regularization under different data augmentations.

However, most of the existing methods of improving data efficiency focus on classification setup while rare attention has been paid to the other side of the coin, *i.e.* deep regression which usually requires more human effort or expert knowledge to labeling. Moreover, due to the intrinsic difference between categorical and continuous label space, the methods based on the cluster or low-density separation assumptions (Lee, 2013; Grandvalet & Bengio, 2005) for data-efficient classification tasks cannot be directly adapted into deep regression. Meanwhile, existing data-efficient regression methods adopt k-nearest neighbor (kNN) (Zhou & Li, 2005b; Yu-Feng Li, 2017), decision tree (Levati et al., 2017) or Gaussian Process (Srijith et al., 2013) as regressors on a fixed and *shallow* feature space, causing them difficult to be extended into deep learning problems.

To develop a general data-efficient deep learning method for both classification and regression setups, we first delved into the existing methods and found that they can be briefly grouped into two categories: 1) Encourage invariance to *data stochasticity* with consistency regularization to make the predictions of the model invariant to local input perturbations, such as Π-model (Laine

---

*Correspondence to: Mingsheng Long (mingsheng@tsinghua.edu.cn)

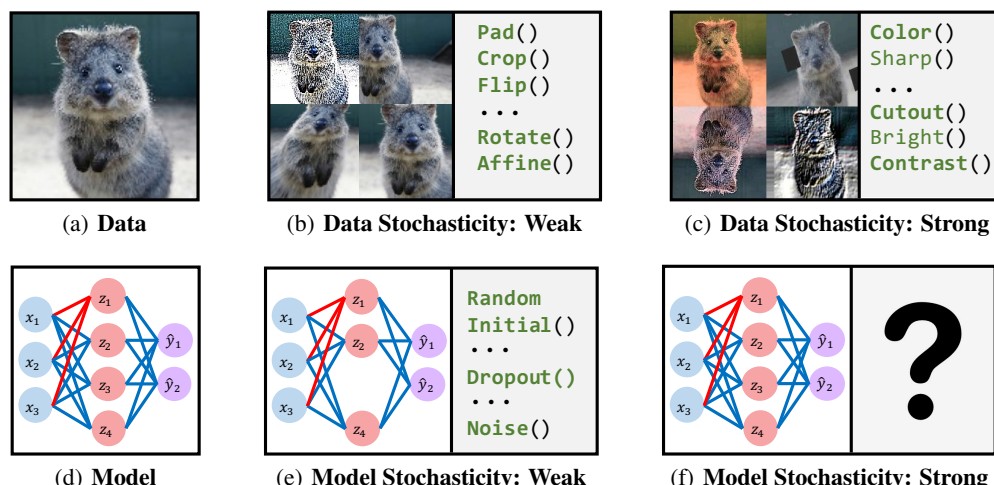

Figure 1: Comparisons among data-efficient techniques. To enhance the invariance to data stochasticity, strong data augmentation has been proposed. Similarly, can we propose a novel approach to further enhance the invariance to model stochasticity?

& Aila, 2017), UDA (Xie et al., 2020) and FixMatch (Sohn et al., 2020); 2) Encourage invariance to *model stochasticity* with difference penalty for predictions of models generated from different dropout (Laine & Aila, 2017) or initialization (Zhou & Li, 2005b), as well as models exponentially averaged from history models, such as Mean Teacher (Tarvainen & Valpola, 2017).

Through the success of these above two strategies of consistency regularization that encourages invariance under data and model transformations respectively, it is intuitive to draw a conclusion that *the invariance to stochasticity matters for improving data efficiency in deep learning*. To take the power of both worlds, we propose a novel $\chi$-Model by simultaneously encouraging the invariance to data stochasticity and model stochasticity. First, instead of the weak augmentations (*e.g.*, flip and crop) adopted in $\Pi$-model, we utilize the strong augmentations (*e.g.*, cutout and contrast) adopted in FixMatch (Sohn et al., 2020) to enhance invariance to data stochasticity, as shown in Figure 1 with some example images tailored from Jung et al. (2020). A natural question arises: Can we further enhance the invariance to model stochasticity similar to that of data stochasticity? This paper gives a positive answer by introducing a minimax game between the feature extractor and task-specific heads. Compared to the manually designed strategy of adding a dropout layer, this novel approach directly optimizes a minimax loss function in the hypothesis space. By maximizing the inconsistency between task-specific heads, more diverse learners are generated to further enhance the invariance to model stochasticity and thus fully explore the intrinsic structure of unlabeled data.

In short, our contributions can be summarized as follows:

- We propose the $\chi$-Model that jointly encourages the invariance to *data stochasticity* and *model stochasticity* to improve data efficiency for both classification and regression setups.
- We make the $\chi$-Model play a minimax game between the feature extractor and task-specific heads to further enhance invariance to model stochasticity.
- Extensive experiments verify the superiority of the $\chi$-Model among various tasks, from an age estimation task to a dense-value prediction task of keypoint localization, a 2D synthetic and a 3D realistic dataset, as well as a multi-category object recognition task.

## 2 RELATED WORK

### 2.1 DATA-EFFICENT CLASSIFICATION

In absence of abundant labeled data, it is reasonable to further explore the additional unlabeled data. A popular approach among these algorithms is Pseudo Labeling (Lee, 2013) which leverages the

model itself to generate labels for unlabeled data and uses generated labels for training. Besides Pseudo Labeling, there is another family of algorithms under the umbrella of "self-training", which has received much attention both empirically (Sohn et al., 2020) and theoretically (Wei et al., 2021). They either enforce stability of predictions under different data augmentations (Tarvainen & Valpola, 2017; Xie et al., 2020; Sohn et al., 2020) (a.k.a. input consistency regularization) or fit the unlabeled data on its predictions generated by a previously learned model (Lee, 2013; Chen et al., 2020b). Further, Co-Training (Blum & Mitchell, 1998), Deep Co-Training Qiao et al. (2018) and Tri-Training (Zhou & Li, 2005a) improve data efficiency from an interesting perspective of different views of classifiers. MixMatch (Berthelot et al., 2019), ReMixMatch (Berthelot et al., 2020) and UDA (Xie et al., 2020) reveal the crucial role of noise produced by advanced data augmentation methods. MCD is also a minimax model originally proposed for domain adaptation. However, it focuses on enhancing invariance to model stochasticity and has a different network architecture with $\chi$-Model. FixMatch (Sohn et al., 2020) uses predictions from weakly-augmented images to supervise the output of strongly augmented data. Meta Pseudo Labels (Pham et al., 2021) further improves data efficiency by making the teacher constantly adapted by the feedback of the student's performance on the labeled dataset. SimCLRv2 (Chen et al., 2020b) first fine-tunes the pre-trained model from the labeled data and then distills on the unlabeled data. Self-Tuning (Wang et al., 2021) introduces a pseudo group contrast (PGC) mechanism but is limited on classification setup. Besides of involving unlabeled data from the same distribution, another promising direction for improving data efficiency is introducing a complementary perspective to further improve data efficiency by introducing a related but different domain (Long et al., 2015; Ganin & Lempitsky, 2015; Long et al., 2017; Saito et al., 2018b; Lee et al., 2019; Zhang et al., 2019; Saito et al., 2018a; 2019). Moreover, various recent methods (van den Oord et al., 2018; He et al., 2020; Wu et al., 2018; Hadsell et al., 2006; Tian et al., 2019; Chen et al., 2020a) improve data efficiency by self-supervised learning. *However, most existing data-efficient methods focus on classification setup while rare attention has been paid to deep regression.*

Table 1: Comparison among various methods for improving data efficiency in deep learning.

| Method | stochasticity | | setup | |
| --- | --- | --- | --- | --- |
| | data | model | classification | regression |
| Pseudo Label (Lee, 2013) | weak | ✗ | ✓ | ✗ |
| Entropy (Grandvalet & Bengio, 2005) | ✗ | ✗ | ✓ | ✗ |
| VAT (Miyato et al., 2016) | weak | ✗ | ✓ | ✓ |
| Π-model (Laine & Aila, 2017) | weak | weak | ✓ | ✓ |
| Data Distillation (Radosavovic et al., 2017) | weak | ✗ | ✓ | ✗ |
| Mean Teacher (Tarvainen & Valpola, 2017) | weak | weak | ✓ | ✓ |
| MCD (Saito et al., 2018b) | weak | strong | ✓ | ✓ |
| UDA (Xie et al., 2020) | strong | ✗ | ✓ | ✗ |
| FixMatch (Sohn et al., 2020) | strong | ✗ | ✓ | ✗ |
| Self-Tuning (Wang et al., 2021) | strong | ✗ | ✓ | ✗ |
| $\chi$-Model (proposed) | strong | strong | ✓ | ✓ |

## 2.2 DATA-EFFICIENT REGRESSION

Data-efficient regression methods mainly fall into three categories: Co-Training, kernel regression, and graph Laplacian regularization paradigm. COREG (Zhou & Li, 2005b) adopt two regressors (kNNs) learned by different distance metrics and predict unlabeled data on one regressor and using the most promising predictions to train the other one. Transductive Regression (Cortes & Mohri, 2007) exploits local linear regression for unlabeled data, takes those with relatively close neighbors, and trains a kernel regressor to produce the final prediction. Graph Laplacian regularization is a commonly used manifold regularization technique (Belkin et al., 2006). It is reasonable to assume that data points with close input values should have similar output values, thereby regularizing the model output with respect to the unlabeled data. Levati et al. (2017) and Srijith et al. (2013) adopt decision tree and Gaussian Process as regressors respectively. *However, these existing data-efficient regression methods are mainly designed for shallow regression*, which requires closed-form or convex-solvers to solve the problem, causing them not suitable for deep regression problems. The comparison of various methods for improving data efficiency in deep learning is shown in Table 1.

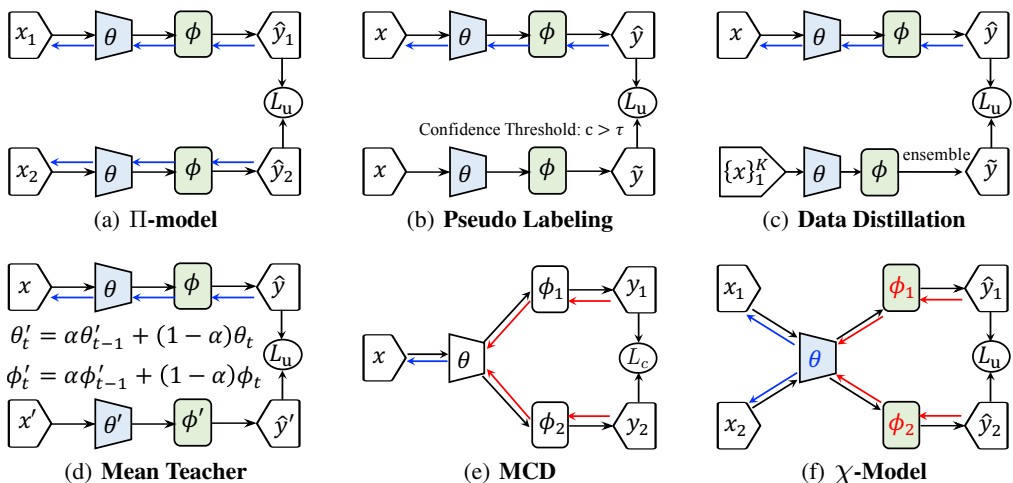

Figure 2: Comparisons among data-efficient learning techniques. (a) **Π-model**: uses a consistency regularization term on predictions of two different augmentations; (b) **Pseudo-Labeling**: leverages the model itself to generate labels for unlabeled data and uses generated labels for training; (c) **Data Distillation**: extends the dual-copies of Π-model into multiple transformations; (d) **Mean Teacher**: maintains the teacher model with an exponential moving average of model weights of the student model and encourages consistency between them. (e) **MCD**: a minimax model originally proposed for domain adaptation that focuses on enhancing invariance to model stochasticity. (f) **χ-Model**: a minimax model for improving data-efficiency by enhancing model stochasticity and data stochasticity.

## 3 PRELIMINARIES

Denote a labeled dataset $\mathcal{L} = \left\{ \left( \boldsymbol{x}_i^L, \boldsymbol{y}_i^L \right) \right\}_{i=1}^{n_L}$ with $n_L$ samples $\left( \boldsymbol{x}_i^L, \boldsymbol{y}_i^L \right)$, and an unlabeled dataset $\mathcal{U} = \left\{ \left( \boldsymbol{x}_i^U \right) \right\}_{i=1}^{n_U}$ with $n_U$ unlabeled samples. Usually, the size $n_L$ of $\mathcal{L}$ is much smaller than that $n_U$ of $\mathcal{U}$ and we define the the label ratio as $n_L/(n_L + n_U)$. Denote $\theta$ the feature generator network, and $\phi$ the successive task-specific head network. We aim at improving data efficiency in deep learning by fully exploring the labeled and unlabeled data from the perspective of stochasticity.

### 3.1 INVARIANCE TO DATA STOCHASTICITY

Data-efficient methods with the insight of *invariance to data stochasticity* aim at making the predictions of the model invariant to local input perturbations by a consistency regularization term. As shown in Figure 2(a), Π-model (Laine & Aila, 2017; Sajjadi et al., 2016) first generates two examples with different stochastic data augmentations and then introduces a loss term to minimize the distance between their predictions. With the strategy of invariance to data stochasticity, Π-model is believed to augment the model with information about the intrinsic structure ("manifold") of $\mathcal{U}$, avoiding overfitting to the labeled data $\mathcal{L}$. Realizing that training a model on its own predictions (*e.g.*, Π-model) often provides no meaningful information, Data Distillation (Radosavovic et al., 2017) extends the dual-copies of Π-model to multiple transformations as shown in Figure 2(c). It first generates an ensembled prediction for each unlabeled input from the predictions of a single model run on different transformations (*e.g.*, flipping and scaling), and then uses it to guide the training of this input. In summary, the training loss of invariance to data stochasticity can be formalized as

$$\min_{\theta, \phi} L_{\text{data}}(\boldsymbol{x}, \mathcal{U}) = \mathbb{E}_{\boldsymbol{x}_i \in \mathcal{U}} \, \ell \left( (\phi \circ \theta)(\text{aug}_1(\boldsymbol{x}_i)), \, (\phi \circ \theta)(\text{aug}_2(\boldsymbol{x}_i)) \right), \tag{1}$$

where $\ell(\cdot, \cdot)$ is a proper loss function for the target task. For clarity, we focus on a particular data example $\boldsymbol{x}_i$ here and the superscript $U$ is omitted. Recently, instead of the weak augmentations (*e.g.*, flip and crop) adopted in Π-model, many strong augmentations (*e.g.*, cutout and contrast) are adopted in FixMatch (Sohn et al., 2020) to further enhance invariance to data stochasticity.

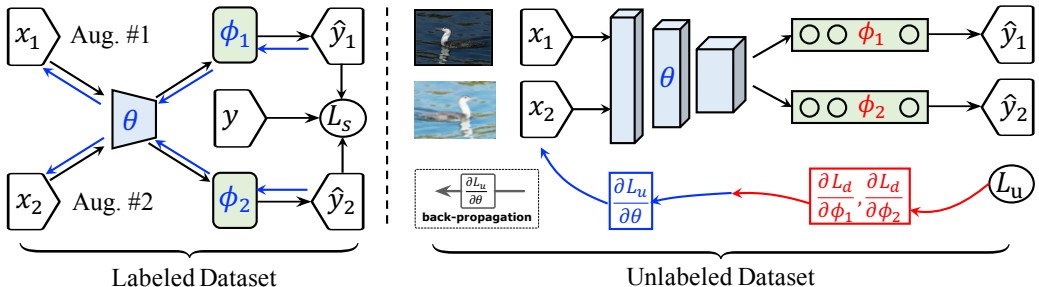

Figure 3: The network architecture of $\chi$-Model on the labeled and unlabeled dataset.

## 3.2 Invariance to Model Stochasticity

Another strategy encourages invariance to *model stochasticity* with difference penalty for predictions of models generated from different dropout (Laine & Aila, 2017) or network initialization (Zhou & Li, 2005b), as well as models exponentially averaged from history models. For example, Mean Teacher (Tarvainen & Valpola, 2017) uses the prediction of an unlabeled sample from the teacher model to guide the learning of a student model as

$$\min_{\theta,\phi} L_{\text{model}}(\boldsymbol{x}, \mathcal{U}) = \mathbb{E}_{\boldsymbol{x}_i \in \mathcal{U}} \, \ell \left( (\phi_t \circ \theta_t)(\boldsymbol{x}_i), \, (\phi'_{t-1} \circ \theta'_{t-1})(\boldsymbol{x}_i) \right), \tag{2}$$

where $(\phi_t \circ \theta_t)$, $(\phi'_{t-1} \circ \theta'_{t-1})$ are the current model and the exponential moving averaged model respectively. Note that $\phi'_t = \alpha\phi'_{t-1} + (1-\alpha)\phi_t, \theta'_t = \alpha\theta'_{t-1} + (1-\alpha)\theta_t$ where $\alpha$ is a smoothing coefficient hyperparameter. Similarly, the strategy of the invariance to model stochasticity tries to improve the quality of the teacher prediction to guide the training of the student model.

## 4 Method

Through the success of the above strategies, it is intuitive that *the invariance to stochasticity matters for improving data efficiency*. To this end, we propose a novel $\chi$-Model with two new designs: 1) Simultaneously encourage the invariance to data stochasticity and model stochasticity, detailed in Section 4.1 and 2) Enhance model stochasticity via a minimax model, detailed in Section 4.2.

### 4.1 Data Stochasticity meets Model Stochasticity

To take the power of both worlds, we propose a novel $\chi$-Model by simultaneously encouraging the invariance to data stochasticity and model stochasticity for improving data efficiency in deep learning. Given an input $\boldsymbol{x}_i$, we first transform it with two different data augmentations as $\text{aug}_1(\boldsymbol{x}_i)$ and $\text{aug}_2(\boldsymbol{x}_i)$ respectively. After that, they will pass the same feature extractor $\theta$ and two different task-specific heads $\phi_1$ and $\phi_2$ to attain predictions $\widehat{\boldsymbol{y}}_{i,1}$ and $\widehat{\boldsymbol{y}}_{i,2}$ with both data stochasticity and model stochasticity as

$$\begin{aligned} \widehat{\boldsymbol{y}}_{i,1} &= (\phi_1 \circ \theta)(\text{aug}_1(\boldsymbol{x}_i)) \\ \widehat{\boldsymbol{y}}_{i,2} &= (\phi_2 \circ \theta)(\text{aug}_2(\boldsymbol{x}_i)), \end{aligned} \tag{3}$$

where $\phi_1$ and $\phi_2$ are two randomly initilized heads with different parameters. For each example $\boldsymbol{x}_i$ in the labeled dataset $\mathcal{L} = \left\{ \left( \boldsymbol{x}_i^L, \boldsymbol{y}_i^L \right) \right\}_{i=1}^{n_L}$, we focus on a particular data example $\boldsymbol{x}_i^L$ and omit superscript $L$. When the invariance to data stochasticity meets with the invariance to model stochasticity, the supervised learning loss on the labeled data $\mathcal{L}$ can be formalized as

$$L_s(\boldsymbol{x}, \mathcal{L}) = \mathbb{E}_{\boldsymbol{x}_i \in \mathcal{L}} \, \ell_s \left( \widehat{\boldsymbol{y}}_{i,1}, \, \boldsymbol{y}_i \right) + \ell_s \left( \widehat{\boldsymbol{y}}_{i,2}, \, \boldsymbol{y}_i \right), \tag{4}$$

where $\ell_s(\cdot, \cdot)$ is a proper loss function for the target task, *e.g.* L1 or L2 loss for regression tasks and cross-entropy loss for classification tasks. Similarly, a direct form of unsupervised loss is

$$L_u(\boldsymbol{x}, \mathcal{U}) = \mathbb{E}_{\boldsymbol{x}_i \in \mathcal{U}} \, \ell_u \left[ (\phi_1 \circ \theta)(\text{aug}_1(\boldsymbol{x}_i)), \, (\phi_2 \circ \theta)(\text{aug}_2(\boldsymbol{x}_i)) \right], \tag{5}$$

where $\ell_u[\cdot,\cdot]$ is a proper loss function to quantify the difference of these predictions, *e.g.* L1 or L2 loss for regression tasks and the symmetric Kullback–Leibler divergence for classification tasks. Different from the strategy that only applies data stochasticity or model stochasticity, this strategy will greatly increase the invariance to stochasticity by unifying the benefit of both worlds.

Note that, instead of the weak augmentations (*e.g.*, flip and crop) adopted in $\Pi$-model, we utilize the strong augmentations (*e.g.*, cutout and contrast) adopted in FixMatch (Sohn et al., 2020) to enhance invariance to data stochasticity, as shown in Figure 1. A natural question arises: Can we further enhance the invariance to model stochasticity similar to that of data stochasticity?

## 4.2 ENHANCE MODEL STOCHASTICITY VIA A MINIMAX MODEL

By minimizing $L_s(\boldsymbol{x}, \mathcal{L})$ on the labeled data and $L_u(\boldsymbol{x}, \mathcal{U})$ on the unlabeled data using a proper trade-off hyper-parameter $\eta$, $L_\mathrm{s} + \eta L_\mathrm{u}$ is an intuitive and effective way to simultaneously encourage the invariance to data stochasticity with model stochasticity. However, since the dropout or random initialization is a kind of weak stochasticity, only posing a minimization loss function between predictions of two task-specific heads may make them generate similar outputs, causing a degeneration problem and providing little meaningful information. To further enhance model stochasticity, we propose *a minimax game* between the feature extractor and task-specific heads as

$$
\begin{aligned}
\widehat{\theta} &= \arg\min_{\theta} \ \ L_s(\boldsymbol{x}, \mathcal{L}) + \eta L_u(\boldsymbol{x}, \mathcal{U}), \\
(\widehat{\phi}_1, \widehat{\phi}_2) &= \arg\min_{\phi_1, \phi_2} \ \ L_s(\boldsymbol{x}, \mathcal{L}) - \eta L_u(\boldsymbol{x}, \mathcal{U}),
\end{aligned}
\tag{6}
$$

In this minimax game, the inconsistency between the predictions of $\phi_1$ and $\phi_2$ on unlabeled data is maximized while the feature extractor is designed to make their predictions as similar as possible. As shown in Figure 3, the blue lines denote the back-propagation of the minimization of loss function while the red ones denote the maximization ones. In the implementation, we can update all parameters simultaneously by multiplying the gradient of the feature extractor $\theta$ with respect to the loss on unlabeled data by a certain negative constant during the backpropagation-based training.

By this minimax strategy, the task-specific heads are believed to capture meaningful information from complementary perspectives to avoid the degeneration problem. Providing more diverse information, the minimax model can further enhance the invariance to model stochasticity. Compared to the manually designed strategy of adding a dropout layer to models or just applying random initialization, this novel approach directly optimizes a minimax loss function in the hypothesis space to fully explore the intrinsic structure of unlabeled data. In this way, $\chi$-Model further enhances invariance to stochasticity from two comprehensive perspectives: data and model.

The comparison between $\chi$-Model with other baselines is summarized in Figure 2. As shown in Figure 2(f), $\mathrm{aug}_1(\boldsymbol{x}_i)$ and $\mathrm{aug}_2(\boldsymbol{x}_i)$ can be regarded as the corners of "$\chi$" on the upper left and lower left respectively, and $\phi_1$ and $\phi_2$ can be denoted by the corners on the upper right and lower right respectively. Inspired by the name of $\Pi$-model that has a shape of "$\Pi$", the proposed strategy with both data stochasticity and model stochasticity is more like a "$\chi$", so we call it $\chi$-Model.

## 5 EXPERIMENTS ON REGRESSION TASKS

### 5.1 2D SYNTHETIC DATASET: *dSprites-Scream*

**dSprites** (Higgins et al., 2017) is a standard 2D synthetic dataset consisting of three subsets each with $737,280$ images: *Color*, *Noisy* and *Scream*, from which we select the most difficilt one *Scream* as the dataset. Example images of *Scream* are shown in Figure 6(a). In each image, there are 5 factors of variations each with a definite value, detailed in Table 7. We adopt position X and position Y and scale as deep regression tasks while excluding the tasks of classification and orientation regression.

*Since recently proposed methods such as UDA, FixMatch, and Self-Tuning adopt pseudo-labels that cannot be exactly defined in deep regression, we did not report their results and these methods are omitted in the following tables.* As shown in Table 2, $\chi$-Model achieves the best performance across various label ratios from $1\%$ to $50\%$ over all baselines, evaluated on the commonly-used MAE measure. It is noteworthy that $\chi$-Model achieves a MAE of $0.092$ provided with only $5\%$ labels,

Table 2: MAE (↓) on tasks of Position X, Position Y and Scale in *dSprites-Scream* (ResNet-18).

| Label Ratio | 1% | | | | 5% | | | | 20% | | | | 50% | | | |
|---|---|---|---|---|---|---|---|---|---|---|---|---|---|---|---|---|
| Method | Scale | X | Y | **All** | Scale | X | Y | **All** | Scale | X | Y | **All** | Scale | X | Y | **All** |
| Labeled Only | .130 | .073 | .075 | .277 | .072 | .036 | .035 | .144 | .051 | .030 | .028 | .108 | .046 | .026 | .025 | .097 |
| VAT (Miyato et al., 2016) | .067 | .042 | .038 | .147 | .046 | .028 | .034 | .109 | .045 | .024 | .029 | .098 | .037 | .027 | .020 | .084 |
| Π-model (Laine & Aila, 2017) | .084 | .035 | .035 | .154 | .058 | .031 | .025 | .114 | .045 | .024 | .023 | .092 | .040 | .021 | .021 | .082 |
| Data Distillation (Radosavovic et al., 2017) | .066 | .039 | .033 | .138 | .045 | .027 | .031 | .104 | .043 | .023 | .026 | .092 | .037 | .023 | .021 | .081 |
| Mean Teacher (Tarvainen & Valpola, 2017) | .062 | .035 | .037 | .134 | .045 | .024 | .033 | .103 | .042 | .023 | .024 | .089 | .038 | .021 | .020 | .079 |
| UDA (Xie et al., 2020) | – | – | – | – | – | – | – | – | – | – | – | – | – | – | – | – |
| FixMatch (Sohn et al., 2020) | – | – | – | – | – | – | – | – | – | – | – | – | – | – | – | – |
| Self-Tuning (Wang et al., 2021) | – | – | – | – | – | – | – | – | – | – | – | – | – | – | – | – |
| $\chi$-Model (w/o minimax) | .080 | **.021** | .024 | .125 | .044 | .029 | .028 | .101 | .040 | .017 | .021 | .077 | **.030** | .027 | .018 | .074 |
| $\chi$-Model (w/o data aug.) | .074 | .025 | **.023** | .119 | .045 | .026 | **.022** | .093 | .037 | .019 | **.017** | .073 | .038 | .018 | **.017** | .074 |
| $\chi$-Model (ours) | **.061** | .030 | .024 | **.115** | **.044** | **.023** | .025 | **.092** | **.037** | **.014** | .021 | **.072** | .032 | **.018** | .018 | **.068** |

which is even lower than that of the labeled-only method (0.097) using 50% labels, indicating a large improvement of data efficiency (10×). Meanwhile, the ablation studies in Table 2 and other ablation studies, verify the effectiveness of both minimax strategy and data augmentation.

## 5.2 3D REALISTIC DATASET: *MPI3D-Realistic*

**MPI3D** (Gondal et al., 2019) is a simulation-to-real dataset for 3D object. It has three subsets: *Toy*, *Realistic* and *Real*, in which each contains $1,036,800$ images. Here, since we have used the synthetic dataset above, we select the *Realistic* subset here to demonstrate the performance of $\chi$-Model. Example images of *Realistic* are shown in Figure 6(b). In each image, there are 7 factors of variations each with a definite value, detailed in Table 8. In *MPI3D-Realistic*, there are two factors that can be employed for regression tasks: Horizontal Axis (a rotation about a vertical axis at the base) and Vertical Axis (a second rotation about a horizontal axis). The goal of this task is to predict the value of the Horizontal Axis and the Vertical Axis for each image via less labeled data.

Table 3: MAE (↓) on tasks of Horizontal Axis and Vertical Axis in *MPI3D-realistic* (ResNet-18).

| Label Ratio | 1% | | | 5% | | | 20% | | | 50% | | |
|---|---|---|---|---|---|---|---|---|---|---|---|---|
| Method | Hor. | Vert. | all | Hor. | Vert. | all | Hor. | Vert. | all | Hor. | Vert. | all |
| Labeled Only | .163 | .112 | .275 | .075 | .042 | .117 | .034 | .030 | .064 | .023 | .019 | .042 |
| VAT (Miyato et al., 2016) | .102 | .087 | .189 | .061 | .048 | .109 | .033 | .028 | .061 | .025 | .021 | .046 |
| Π-model (Laine & Aila, 2017) | .099 | .086 | .185 | .051 | .041 | .092 | .031 | .024 | .055 | .023 | .017 | .040 |
| Data Distillation (Radosavovic et al., 2017) | .095 | .092 | .187 | .053 | .046 | .099 | .029 | .024 | .053 | .023 | .021 | .044 |
| Mean Teacher (Tarvainen & Valpola, 2017) | .091 | .081 | .172 | .045 | .036 | .081 | .031 | .028 | .059 | .024 | .018 | .042 |
| $\chi$-Model (ours) | **.087** | **.068** | **.155** | **.036** | **.024** | **.060** | **.024** | **.018** | **.042** | **.021** | **.016** | **.037** |

Following the previous setup, we evaluate our $\chi$-Model on various label ratios including 1%, 5%, 20% and 50%. As shown in Table 3, $\chi$-Model beats the strong competitors of data stochasticity and model stochasticity-based methods by a large margin when evaluated on the MAE measure. Note that, some data-efficient methods underperform than the vanilla label-only model (*e.g.* Data Distillation (0.044) and VAT (0.046) achieves a higher MAE than label-only method (0.042) when provided with a label ratio of 50%), while our $\chi$-Model always benefits from exploring the unlabeled data.

## 5.3 AGE ESTIMATION DATASET: *IMDB-WIKI*

**IMDB-WIKI**(Rasmus Rothe, 2016) is a face dataset with age and gender labels, which can be used for age estimation. It contains $523.0K$ face images and the corresponding ages. After splitting, there are $191.5K$ images for training and $11.0K$ images for validation and testing. For data-efficient deep regression, we construct 5 subsets with different label ratios including 1%, 5%, 10%, 20% and 50%. For evaluation metrics, we use common metrics for regression, such as the mean-average-error (MAE) and another metric called error Geometric Mean (GM) (Yang et al., 2021) which is defined as $(\Pi_{i=1}^{n} e_i)^{\frac{1}{n}}$ for better prediction fairness, where $e_i$ is the prediction error of each sample $i$.

As reported in Table 4, no matter evaluated by MAE or GM, $\chi$-Model achieves the best performance across various label ratios from 1% to 50% over all baselines. It is noteworthy that the value of the

Table 4: MAE ($\downarrow$) and GM ($\downarrow$) on task of age estimation in *IMDB-WIKI* (ResNet-50).

| Label Ratio | 1% | | 5% | | 10% | | 20% | | 50% | |
|---|---|---|---|---|---|---|---|---|---|---|
| Method | MAE | GM | MAE | GM | MAE | GM | MAE | GM | MAE | GM |
| Labeled Only | 17.72 | 12.69 | 14.79 | 9.93 | 13.70 | 9.05 | 11.38 | 7.16 | 9.53 | 5.74 |
| VAT (Miyato et al., 2016) | 13.02 | 8.55 | 12.09 | 7.73 | 11.06 | 6.74 | 10.43 | 6.49 | 8.38 | 4.88 |
| Π-model (Laine & Aila, 2017) | 12.99 | 8.65 | 11.92 | 7.69 | 10.90 | 6.61 | 10.12 | 6.42 | 8.36 | 4.80 |
| Co-Training (Zhou & Li, 2005b) | 13.46 | 8.74 | 12.63 | 7.93 | 11.28 | 6.78 | 10.78 | 6.58 | 8.61 | 4.99 |
| Data Distillation (Radosavovic et al., 2017) | 13.12 | 8.62 | 12.07 | 7.71 | 11.02 | 6.71 | 10.37 | 6.43 | 8.33 | 4.84 |
| Mean Teacher (Tarvainen & Valpola, 2017) | 12.94 | 8.45 | 12.01 | 7.64 | 10.91 | 6.62 | 10.29 | 6.35 | 8.31 | 4.78 |
| $\chi$-**Model** (ours) | **12.75** | **8.32** | **11.55** | **7.33** | **10.57** | **6.29** | **9.89** | **5.98** | **8.06** | **4.59** |

unlabeled data is highlighted in this dataset since all baselines that try to explore the intrinsic structure of unlabeled data work much better than the vanilla one: labeled-only method.

## 5.4 KEYPOINT LOCALIZATION DATASET: *Hand-3D-Studio*

Hand-3D-Studio (H3D) (Zhao et al., 2020) is a real-world dataset containing $22k$ frames in total. These frames are sampled from videos consisting of hand images of 10 persons with different genders and skin colors. We use the Percentage of Correct Keypoints (PCK) as the measure of evaluation. A keypoint prediction is considered correct if the distance between it with the ground truth is less than a fraction $\alpha = 0.05$ of the image size. PCK@0.05 means the percentage of keypoints that can be considered as correct over all keypoints.

As shown in Table 5, most baselines that explore unlabeled data show competitive performance and the proposed $\chi$-Model still consistently improves over these baselines. For example, when provided with 1% labels, $\chi$-Model achieves an average PCK of 73.6, which has an absolute improvement of 3.4 over the strongest baseline.

Table 5: PCK@0.05 ($\uparrow$) on task of keypoint localization in *Hand-3D-Studio* (ResNet-101).

| Label Ratio | 1% | | | | | 5% | | | | |
|---|---|---|---|---|---|---|---|---|---|---|
| Method | MCP | PIP | DIP | Fingertip | Avg | MCP | PIP | DIP | Fingertip | Avg |
| Labeled Only (Xiao et al., 2018) | 72.3 | 70.3 | 66.3 | 62.8 | 68.5 | 90.5 | 86.7 | 82.0 | 76.3 | 84.4 |
| VAT (Miyato et al., 2016) | 76.2 | 71.2 | 66.0 | 61.6 | 69.6 | 91.2 | 87.9 | 83.1 | 78.5 | 85.6 |
| Π-model (Laine & Aila, 2017) | 76.1 | 70.3 | 65.6 | 61.6 | 69.2 | 90.1 | 86.4 | 82.0 | 76.8 | 84.3 |
| Data Distillation (Radosavovic et al., 2017) | 75.8 | 70.1 | 65.8 | 62.0 | 69.3 | 90.9 | 88.9 | 84.7 | 79.6 | 86.3 |
| Mean Teacher (Tarvainen & Valpola, 2017) | 76.5 | 71.7 | 66.7 | 62.7 | 70.2 | 91.8 | 89.0 | 84.5 | 79.6 | 86.6 |
| $\chi$-**Model** (w/o minimax) | 79.2 | 74.6 | 69.2 | 63.6 | 72.3 | 91.3 | 88.5 | 83.5 | 78.7 | 85.9 |
| $\chi$-**Model** (w/o data aug.) | 79.4 | 75.2 | 70.0 | 64.6 | 72.9 | 91.8 | 89.0 | 84.5 | 79.6 | 86.6 |
| $\chi$-**Model** (ours) | **79.8** | **75.6** | **70.7** | **65.5** | **73.6** | **92.3** | **89.4** | **85.0** | **80.8** | **87.2** |

## 6 EXPERIMENTS ON CLASSIFICATION TASKS

We adopt the most difficult *CIFAR-100* dataset (Krizhevsky, 2009) with 100 categories among the famous data-efficient classification benchmarks including *CIFAR-100*, *CIFAR-10*, *SVHN*, and *STL-10*, where the last three datasets have only 10 categories. As shown in Figure 6, $\chi$-Model not only performs well on deep regression tasks but also outperforms other baselines on classification tasks. Note that, MCD with strong data augmentation still underperforms $\chi$-Model, since the dual branch architecture of $\chi$-Model enables itself to further enhance data stochasticity as compared in Figure 2.

## 7 INSIGHT ANALYSIS

**An Analysis on Two-Moon** We use the classical Two-Moon dataset with *scikit-learn* (Pedregosa et al., 2011) to visualize decision boundaries of various data-efficient learning methods on a binary *classification* task with only 5 labels per class (denoted by "+"). As shown in Figure 4, from the

Table 6: Error rates (%) ↓ of classification on *CIFAR-100* (WRN-28-8).

| Method | 400 labels | 2500 labels | 10000 labels |
|---|---|---|---|
| Pseudo-Labeling (Lee, 2013) | - | 57.38±0.46 | 36.21±0.19 |
| MC Dropout (Gal & Ghahramani, 2016) | - | 58.27±0.54 | 38.36±0.19 |
| Deep Co-Training (Qiao et al., 2018) | - | 53.38±0.61 | 34.63±0.14 |
| Π-Model (Laine & Aila, 2017) | - | 57.25±0.48 | 37.88±0.11 |
| MME (Saito et al., 2019) | - | 47.40±1.75 | 32.54±0.81 |
| Mean Teacher (Tarvainen & Valpola, 2017) | - | 53.91±0.57 | 35.83±0.24 |
| MCD (weak aug.) (Saito et al., 2018b) | - | 36.17±0.33 | 28.47±0.38 |
| MCD (strong aug.) (Saito et al., 2018b) | - | 29.59±0.41 | 23.10±0.23 |
| MixMatch (Berthelot et al., 2019) | 67.61±1.32 | 39.94±0.37 | 28.31±0.33 |
| UDA (Xie et al., 2020) | 59.28±0.88 | 33.13±0.22 | 24.50±0.25 |
| ReMixMatch (Berthelot et al., 2020) | **44.28**±2.06 | 27.43±0.31 | 23.03±0.56 |
| FixMatch (Sohn et al., 2020) | 48.85±1.75 | 28.29±0.11 | 22.60±0.12 |
| Meta Pseudo Labels (Pham et al., 2021) | 48.18±1.29 | 27.31±0.24 | 22.02±0.18 |
| Self-Tuning (Wang et al., 2021) | 54.74±0.35 | 42.08±0.43 | 21.75±0.27 |
| $\chi$-**Model** | 47.21±1.54 | **27.11**±0.65 | **20.98**±0.33 |

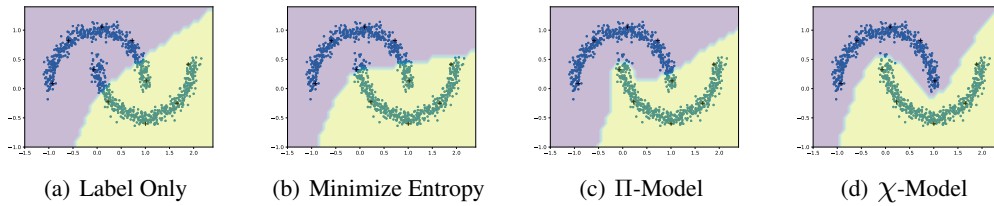

(a) Label Only     (b) Minimize Entropy     (c) Π-Model     (d) $\chi$-Model

Figure 4: Decision boundaries of various data-efficient learning methods on Two-Moon.

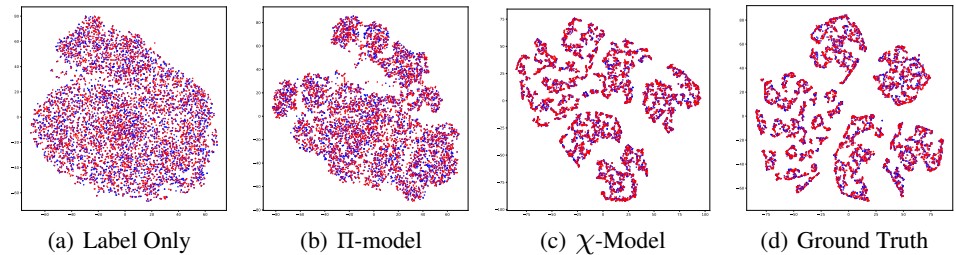

(a) Label Only     (b) Π-model     (c) $\chi$-Model     (d) Ground Truth

Figure 5: *t*-SNE features on dSprites-Scream. (Red: labeled samples; Blue: unlabeled samples).

label-only method to $\chi$-Model, the decision boundary is more and more discriminable and a more accurate hyperplane between two classes is learned.

**Feature Visualization via *t*-SNE**   We use the *t*-SNE tool (Maaten & Hinton, 2008) to visualize the features of several models on a *regression* task named dSprites-Scream with 5% labels. As shown in Figure 5, from the label-only method to $\chi$-Model, the intrinsic structure ("manifold") of the unlabeled data is more discriminable and much more similar to that of the fully-supervised ground-truth one.

## 8   CONCLUSION

We propose the $\chi$-Model, which unifies the invariance to *data stochasticity* and *model stochasticity* to take the power of both worlds with a form of "$\chi$". We make the $\chi$-Model play a minimax game between the feature extractor and task-specific heads to further enhance invariance to stochasticity. Extensive experiments demonstrate the superior performance of the $\chi$-Model among various tasks.

## ACKNOWLEDGEMENTS

This work was supported by the National Megaproject for New Generation AI (2020AAA0109201), National Natural Science Foundation of China (62022050 and 62021002), Beijing Nova Program (Z201100006820041), BNRist Innovation Fund (BNR2021RC01002) and Kuaishou Technology Fund. This work was done when Ximei was also an intern at Kuaishou Technology and supervised by Guoxin Zhang and Pengfei Wan.

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

## A    EXPERIMENT DETAILS

We empirically evaluate $\chi$-Model in several dimensions:

- **Task Variety**: four regression tasks with various dataset scales including single-value prediction task of age estimation, to dense-value prediction ones of keypoint localization, as well as a 2D synthetic dataset and a 3D realistic dataset. We further include a multi-category object recognition task to verify the effectiveness of $\chi$-Model on classfication setup.

- **Label Proportion**: the proportion of labeled dataset ranging from $1\%$ to $50\%$ following the popular protocol of data-efficient deep learning.

- **Pre-trained Models**: mainstream pre-trained models are adopted including ResNet-18, ResNet-50 and ResNet-101 (He et al., 2016).

**Baselines**    We compared $\chi$-Model against three types of baselines: (1) **Data Stochasticity**: $\Pi$-model (Laine & Aila, 2017), VAT (Miyato et al., 2016), and Data Distillation (Radosavovic et al., 2017); (2) **Model Stochasticity**: Mean Teacher (Tarvainen & Valpola, 2017); (3) **Label-Only Models**: a task-specific model trained on only the labeled data is provided. For example, Simple Baseline (Xiao et al., 2018) with a ResNet101 pre-trained backbone trained on only the labeled data is used a strong baseline for keypoint localization.

**Implementation Details**    We use **PyTorch**[1] with Titan V to implement our methods. Models pre-trained on ImageNet such as ResNet-18, ResNet-50, ResNet-101 (He et al., 2016) are used for the backbones of experiments on *dSprites-Scream*, *IMDB-WIKI*, and *Hand-3D-Studio* respectively.

On *dSprites-Scream*, there are 5 factors of variations each with a definite value in each image. Among these factors, there are four factors that can be employed for regression tasks: scale, orientation, position X and position Y. Therefore, we adopt position X and position Y and scale as deep regression tasks while excluding the task of orientation regression. We treat all tasks equally and address there tasks as a multi-task learning problem with a shared backbone and different heads. Labels are all normalized to $[0, 1]$ to eliminate the effects of diverse scale in regression values, since the activation of the regressor is sigmoid whose value range is also $[0, 1]$.

On *MPI3D-Realistic*, there are two factors that can be employed for regression tasks: Horizontal Axis (a rotation about a vertical axis at the base) and Vertical Axis (a second rotation about a horizontal axis), since each object is mounted on the tip of the manipulator and the manipulator has two degrees of freedom. The goal of this data-efficient deep regression task is to train a model to accurately predict the value of the Horizontal Axis and the Vertical Axis for each image via less labeled data.

On *IMDB-WIKI*, following the data pre-process method of a recent work (Yang et al., 2021), we also filter out unqualified images, and manually construct balanced validation and test set over the supported ages. After splitting, there are $191.5K$ images for training and $11.0K$ images for validation and testing. For data-efficient deep regression, we construct 5 subsets with different label ratios including $1\%, 5\%, 10\%, 20\%$ and $50\%$. For evaluation metrics, we use common metrics for regression, such as the mean-average-error (MAE) and another metric called error Geometric Mean (GM) (Yang et al., 2021) which is defined as $(\Pi_{i=1}^{n} e_i)^{\frac{1}{n}}$ for better prediction fairness, where $e_i$ is the prediction error of each sample $i$.

On *Hand-3D-Studio*, we first randomly split the dataset into a testing set with $3.2k$ frames and a training set containing the remaining part. Further, we follow the setup of data-efficient learning and construct subsets with label ratios of $1\%$ and $5\%$. We use the Percentage of Correct Keypoints (PCK) as the measure of evaluation. A keypoint prediction is considered correct if the distance between it with the ground truth is less than a fraction $\alpha = 0.05$ of the image size. PCK@0.05 means the percentage of keypoints that can be considered as correct over all keypoints.

All images are resized to $224 \times 224$. The tradeoff hyperparameter $\eta$ is set as $0.1$ for all tasks unless specified. The learning rates of the heads are set as $10$ times to those of the backbone layers, following the common fine-tuning principle (Yosinski et al., 2014). We adopt the mini-batch SGD with momentum of $0.95$. Code will be made available at https://github.com.

---

[1]`http://pytorch.org`

## B  EXAMPLE IMAGES AND FACTORS OF DSPRITES AND MPI3D

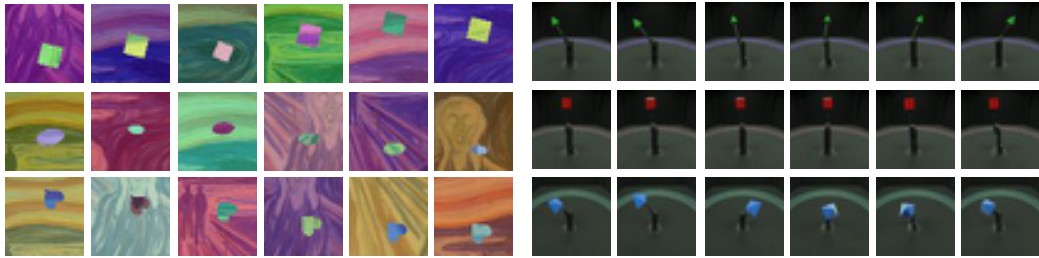

(a) Example images of *dSprites-Scream*    (b) Example images of *MPI3D-Realistic*

Table 7: Factors of variations in *dSpirtes*

| Factor | Possible Values | Task |
|---|---|---|
| Shape | square, ellipse, heart | recognition |
| Scale | 6 values in $[0.5, 1]$ | regression |
| Orientation | 40 values in $[0, 2\pi]$ | regression |
| Position X | 32 values in $[0, 1]$ | regression |
| Position Y | 32 values in $[0, 1]$ | regression |

Table 8: Factors of variations in *MPI3D*

| Factor | Possible Values | Task |
|---|---|---|
| Object Size | small=0, large=1 | recognition |
| Camera Height | top=0, center=1, bottom=2 | recognition |
| B.G. Color | purple=0, green=1, salmon=2 | recognition |
| Horizontal Axis | 40 values in $[0, 1]$ | regression |
| Vertical Axis | 40 values in $[0, 1]$ | regression |

## C  FULL RESULTS OF IMDB-WIKI

**IMDB-WIKI**[2] (Rasmus Rothe, 2016) is a face dataset with age and gender labels, which can be used for age estimation. It contains $523.0K$ face images and the corresponding ages. Following the data pre-process method of a recent work (Yang et al., 2021), we also filter out unqualified images, and manually construct balanced validation and test set over the supported ages. After splitting, there are $191.5K$ images for training and $11.0K$ images for validation and testing. For data-efficient deep regression, we construct 5 subsets with different label ratios including $1\%$, $5\%$, $10\%$, $20\%$ and $50\%$. For evaluation metrics, we use common metrics for regression, such as the mean-average-error (MAE) and another metric called error Geometric Mean (GM) (Yang et al., 2021) which is defined as $(\Pi_{i=1}^{n} e_i)^{\frac{1}{n}}$ for better prediction fairness, where $e_i$ is the prediction error of each sample $i$.

As reported in Table 9, no matter evaluated by MAE or GM, $\chi$-Model achieves the best performance across various label ratios from $1\%$ to $50\%$ over all baselines. It is noteworthy that the value of the unlabeled data is highlighted in this dataset since all baselines that try to explore the intrinsic structure of unlabeled data works much better than the vanilla one: labeled-only method.

Table 9:  MAE ($\downarrow$) and GM ($\downarrow$) on task of age estimation in *IMDB-WIKI* (ResNet-50).

| Label Ratio | 1% | | 5% | | 10% | | 20% | | 50% | |
|---|---|---|---|---|---|---|---|---|---|---|
| Method | MAE | GM | MAE | GM | MAE | GM | MAE | GM | MAE | GM |
| Labeled Only | 17.72 | 12.69 | 14.79 | 9.93 | 13.70 | 9.05 | 11.38 | 7.16 | 9.53 | 5.74 |
| VAT (Miyato et al., 2016) | 13.02 | 8.55 | 12.09 | 7.73 | 11.06 | 6.74 | 10.43 | 6.49 | 8.38 | 4.88 |
| Π-model (Laine & Aila, 2017) | 12.99 | 8.65 | 11.92 | 7.69 | 10.90 | 6.61 | 10.12 | 6.42 | 8.36 | 4.80 |
| Data Distillation (Radosavovic et al., 2017) | 13.12 | 8.62 | 12.07 | 7.71 | 11.02 | 6.71 | 10.37 | 6.43 | 8.33 | 4.84 |
| Mean Teacher (Tarvainen & Valpola, 2017) | 12.94 | 8.45 | 12.01 | 7.64 | 10.91 | 6.62 | 10.29 | 6.35 | 8.31 | 4.78 |
| $\chi$-**Model** (w/o minimax) | 12.87 | 8.51 | 11.91 | 7.60 | 10.73 | 6.55 | 10.00 | 6.06 | 8.22 | 4.59 |
| $\chi$-**Model** (w/o data aug.) | 13.02 | 8.64 | 11.71 | 7.44 | 11.06 | 6.86 | 10.10 | 6.03 | 8.17 | **4.57** |
| $\chi$-**Model** (ours) | **12.75** | **8.32** | **11.55** | **7.33** | **10.57** | **6.29** | **9.89** | **5.98** | **8.06** | 4.59 |

---

[2]https://data.vision.ee.ethz.ch/cvl/rrothe/imdb-wiki/

