# OpenReview forum: "X-model: Improving Data Efficiency in Deep Learning with A Minimax Model"
_ICLR.cc/2022/Conference — ICLR 2022 Poster_

### Official Review · Reviewer_3Tfv · 2021-11-01

**Correctness:** 4
**Technical Novelty And Significance:** 3
**Empirical Novelty And Significance:** 3
**Recommendation:** 8
**Confidence:** 4

**Main Review:**

Strengths:
*  Data-efficient regression is a very interesting direction. Most previous works on data-efficient methods focus only on classification setups. For example, pseudo-labels, which is proven to be one of the most effective methods for semi-supervised classification, cannot be exactly defined in a deep regression problem. This paper sheds some light on this new direction.
* The proposed method seems to be simple but effective.

Weaknesses:
* One missing related work on pseudo labels [1]. It would be great to include some discussion or even add as a baseline in the classification experiments.
* It is unclear whether the proposed method can generalize well to more challenging tasks (for example, depth estimation or larger dataset). Though, I'm satisfied with the current evaluation (from a single-value toy dataset to dense-value prediction datasets).

[1] Pham, Hieu, et al. "Meta pseudo labels." CVPR 2021.


**Summary Of The Paper:**

The paper presents a g data-efficient approach that encourages invariance to both data and model stochasticity that works for both classification and regression tasks. Furthermore, the proposed minimax loss function can specifically enhance invariance to model stochasticity. The extensive experimental results verify that the proposed method is effective.

**Summary Of The Review:**

In general, the proposed method is well-motivated with sufficient experimental comparisons. Most importantly, the paper seems to be the first step toward data-efficient regression leveraging strong stochasticities from both data and model. To this end, I believe this paper has enough merit to be accepted.

---

> ### Author Response · Authors · 2021-11-19
> **Response to Reviewer 3Tfv**
>
> Many thanks for the recognition of our work from Reviewer 3Tfv. We have clarified the questions in the following feedback.
>
> **Q1:** Comparison with Meta Pseudo labels.
>
> Thanks for pointing out this great work. Similar to pseudo-labeling methods such as FixMatch, Meta Pseudo Labels also has a teacher network to generate pseudo labels on unlabeled data to teach a student network. However, the teacher in Meta Pseudo Labels is constantly adapted by the feedback of the student’s performance on the labeled dataset.
> We have included some discussion with Meta Pseudo Labels [1] in the related work section and added it as a baseline in the experiments on Cifar-100. Like FixMatch and $\chi$-model, the same RandAug data augmentation method is applied in Meta Pseudo Label. The results show that $\chi$-model is comparable with Meta Pseudo Labels on the classification setup.
>
> | Method  | 400 labels | 2500 labels | 10000 labels |
> |-----------|----------- |-----------| -----------|
> | FixMatch | 48.85 ± 1.75 | 28.29 ± 0.11 | 22.60 ± 0.12 |
> | Meta Pseudo Labels (w. RandAug) | 48.18 ± 1.29 | 27.31 ± 0.24 | 22.02 ± 0.18|
> | $\chi$-model | 47.21 ± 1.54 | 27.11 ± 0.65 | 20.98 ± 0.33 |
>
> Further, Meta Pseudo Labels also adopts a *pseudo-labeling strategy* and cannot be naturally adapted into *deep regression*, while $\chi$-model is general to both classification and regression setups.
>
>
>
> **Q2:** Whether can $\chi$-model generalize well to more challenging tasks?
>
> Thanks for your suggestion. Thanks to the general design of the $\chi$-model, we hold positive on its performance on more challenging tasks. Meanwhile, extensive experiments have already verified the superiority of the $\chi$-model among various tasks, from a single-value prediction task of age estimation to a dense-value prediction task of keypoint localization, a 2D synthetic, and a 3D realistic dataset, as well as a multi-category object recognition task.
> Due to the time limit, we did not have enough time to conduct experiments on more challenging tasks such as depth estimation or larger datasets. We will report the results on these tasks in the future version of the $\chi$-model.
>
> [1] Pham, Hieu, et al. "Meta pseudo labels." CVPR 2021.

---

### Official Review · Reviewer_DfEj · 2021-11-03

**Correctness:** 3
**Technical Novelty And Significance:** 2
**Empirical Novelty And Significance:** 2
**Recommendation:** 6
**Confidence:** 4

**Main Review:**

## Strong points
Good performance on a wide range of benchmark datasets.
Easy to read, experiments clearly described.
Approach is fit for regression as well.


## Weak points
Two heads which i assume do not share weights are similar to an ensemble and can hence be compared with ensembling based approaches. Even more if comparing with a MC Dropout based ensemble, where the dropout might be applied in the final layers only, hence effectively creating an ensemble with a backbone with shared weights. [4]
To my understanding the proposed approach is the same as if applying random data augmentations and passing through a DNN with dropout sampling twice. Training one model on the label provided by another model is known as co-training [1] and has been applied for Semi-supervised learning (SSL) also recently [2] e.g. on CIFAR100 results, especially w/o data augmentation would aid in judging the contribution of individual components of the proposed approach.
There is a huge amount of SSL literature that would be needed to be taken into account, on guide could be [3] - please check for more recent ones as well.


## Questions
Do both heads of the chi-model have the same architecture and do not share weights?
Are the heads only one-layer or multi-layered architectures?


## Minor comments
Evaluations w/ and w/o data augmentation especially when comparing with other methods would be helpful. At least stating which methods use the same data augmentations.

[1] Blum, Avrim, and Tom Mitchell. "Combining labeled and unlabeled data with co-training." Proceedings of the eleventh annual conference on Computational learning theory. 1998.
[2] Qiao, Siyuan, et al. "Deep co-training for semi-supervised image recognition." Proceedings of the european conference on computer vision (eccv). 2018.
[3] Zhu, Xiaojin Jerry. "Semi-supervised learning literature survey." (2005).
[4] Gal, Yarin, and Zoubin Ghahramani. "Dropout as a bayesian approximation: Representing model uncertainty in deep learning." international conference on machine learning. PMLR, 2016.

**Summary Of The Paper:**

This paper proposes the so called Chi-model, which combines data augmentation with a two-headed network architecture to include model stochasticity. It specifically addresses data efficient learning in regression settings.

**Summary Of The Review:**

Approach seems promising, especially the empirical results. I am lacking a bit of information to set this into relation to previous works as outlined above. Without these details it is hard to judge how much can be attributed to the proposed approach.

---

> ### Author Response · Authors · 2021-11-19
> **Response to Reviewer DfEj**
>
>
> We thank the constructive comments from Reviewer DfEj and provide further information and experiments to verify the contribution of the proposed $\chi$-model.
>
> **Q1:** Comparison with ensemble methods such as MC Dropout [4].
>
> $\chi$-model is very different from ensemble methods such as MC Dropout. The biggest difference is that $\chi$-model proposes a minimax game between the feature extractor and task-specific heads to further enhance the invariance to model stochasticity. In this minimax game, the inconsistency between the predictions of $\phi_1$ and $\phi_2$ on unlabeled data is maximized while the feature extractor is designed to make their predictions as similar as possible.
> On the contrary, the dropout operation that MC Dropout adopts is a kind of weak stochasticity, only posing a minimization loss function between predictions of two task-specific heads may make them generate similar outputs, causing a degeneration problem and providing little meaningful information. The experiments on Cifar-100 as shown below also verify that it underperforms the proposed $\chi$-model.
>
> | Method  | 2500 labels | 10000 labels |
> |-----------|-----------| -----------|
> | MC Dropout | 58.27±0.54 | 38.36±0.19 |
> | Pi-Model | 57.25±0.48 | 37.88±0.11 |
> | $\chi$-model (Ours) | 27.11±0.65  | 20.98±0.33 |
>
>
>
> **Q2:** Comparison with a method that applies random data augmentations and passes through a DNN with dropout sampling twice
>
> Similarly, the biggest difference between $\chi$-model and the DNN fully Dropout method is that $\chi$-model proposes a minimax game between the feature extractor and task-specific heads to further enhance the invariance to the model stochasticity.
> By further introducing model stochasticity through a DNN with dropout sampling twice, the DNN fully Dropout method outperforms the MC Dropout method that applies dropout in the final layers only.
>
> | Method  | 2500 labels | 10000 labels |
> |-----------|-----------| -----------|
> | MC Dropout | 58.27±0.54 | 38.36±0.19 |
> | DNN fully Dropout | 57.38±0.46 | 38.04±0.21 |
> | Pi-Model | 57.25±0.48 | 37.88±0.11 |
> | $\chi$-model (Ours) | 27.11±0.65  | 20.98±0.33 |
>
>
> **Q3:** Comparison with Co-training [1] and Deep Co-training [2]
>
> Since Co-Training is mainly defined in the shallow feature space, we provide the results of Deep Co-Training that further exploits adversarial examples to encourage view difference.
> For a fair comparison, we report the results of Deep Co-Training and $\chi$-model with and without data augmentation.
>
> | Method  | 2500 labels | 10000 labels |
> |-----------|-----------| -----------|
> | Deep Co-Training (w/o data aug.) | 59.01±0.23 | 38.77±0.28 |
> | Deep Co-Training (w/ data aug.) | 53.38±0.61 | 34.63±0.14 |
> | $\chi$-model (w/o data aug.) | 36.17±0.33  | 28.47±0.38 |
> | $\chi$-model (Ours) | 27.11±0.65  | 20.98±0.33 |
>
> Moreover, Co-Training trains one model on the selected label provided by another model and it requires a metric to select the most confident labels. Therefore, another drawback of Co-Training is that it can not be directly applied to regression setup.
> Although an existing data-efficient regression method named COREG  extends Co-Training into regression setup, COREG adopts k-nearest neighbor (kNN) as regressors on a fixed and shallow feature space, causing it difficult to be applied to deep learning problems.

---

> > ### Author Response · Authors · 2021-11-19
> > **Response to Reviewer DfEj**
> >
> > **Q4:** Survey of other SSL literature [3].
> >
> > We also discussed some classical and various recently proposed methods in the Related Work Section, including Meta Pseudo Labels [6], MixMatch [7], UDA [8], ReMixMatch [9], Co-Training [1], Deep Co-Training, Tri-Training [10] to make the survey of SSL literature more comprehensive.
> > Among these methods, most of them have been added as baselines and compared in the Experiments Section in the first draft or in this rebuttal.
> > In summary, these methods either only focus on classification setup, leaving the deep regression problem on the shelf, or just encourage invariance to data or model stochasticity, without adopting a minimax game to further enhance invariance to model stochasticity.
> >
> >
> > **Q5:** Do both heads of the $\chi$-model have the same architecture and do not share weights?
> >
> > Yes, they have the same architecture and do not share weights, since sharing weight between heads will make them generate similar outputs, causing a degeneration problem and providing little meaningful information.
> >
> > **Q6:** Are the heads only one-layer or multi-layered architectures?
> >
> > The architecture of the heads is task-specific. For single-value regression tasks such as age estimation on IMDB-WIKI, it has just one fully-connected layer. For dense-value regression tasks such as keypoint localization on Hand-3D-Studio, several layers of **nn.Conv2d(), nn.BatchNorm2d(), nn.ReLU()** are included, according to the architecture of Simple Baseline [5], a state-of-the-art architecture we adopted to tackle the keypoint localization task.
> >
> > **Q7:** Data augmentation details of other methods.
> >
> > As stated in Table 1 of the original paper, Pseudo Labeling, Entropy Minimization, VAT, Pi-Model, Data Distillation and Mean Teacher adopt the weak data augmentation such as flip and crop, while UDA, FixMatch, and $\chi$-model, as well as the Meta Pseudo Labels mentioned by Reviewer 3Tfv, use the same RandAug data augmentation method.
> >
> >
> > [1] Blum, Avrim, and Tom Mitchell. "Combining labeled and unlabeled data with co-training." Proceedings of the eleventh annual conference on Computational learning theory. 1998.
> >
> > [2] Qiao, Siyuan, et al. "Deep co-training for semi-supervised image recognition." Proceedings of the European conference on computer vision (eccv). 2018.
> >
> > [3] Zhu, Xiaojin Jerry. "Semi-supervised learning literature survey." (2005).
> >
> > [4] Gal, Yarin, and Zoubin Ghahramani. "Dropout as a Bayesian approximation: Representing model uncertainty in deep learning." international conference on machine learning. PMLR, 2016.
> >
> > [5] Bin Xiao, Haiping Wu, and Yichen Wei. Simple baselines for human pose estimation and tracking. ECCV, 2018.
> >
> > [6] Pham, Hieu, et al. "Meta pseudo labels." CVPR 2021.
> >
> > [7] David Berthelot, Nicholas Carlini, Ian J. Goodfellow, Nicolas Papernot, Avital Oliver, and Colin Raffel. Mixmatch: A holistic approach to semi-supervised learning. NeurIPS, 2019.
> >
> > [8] Qizhe Xie, Zihang Dai, Eduard Hovy, Minh-Thang Luong, and Quoc V Le. Unsupervised data augmentation for consistency training. NeurIPS, 2020.
> >
> > [9] David Berthelot, Nicholas Carlini, Ekin D. Cubuk, Alex Kurakin, Kihyuk Sohn, Han Zhang, and Colin Raffel. Remixmatch: Semi-supervised learning with distribution alignment and augmentation anchoring. ICLR, 2020.
> >
> > [10] Zhou, ZH., Li, M. Tri-training: Exploiting unlabeled data using three classifiers. TKDE, 2005.

---

> > > ### Author Response · Authors · 2021-11-25
> > > **Response to Reviewer DfEj**
> > >
> > > Dear Reviewer DfEj,
> > >
> > > Many thanks for your time and efforts in reviewing our paper.
> > >
> > > We kindly remind that we have only one week for the discussion period. We have made an extensive effort to try to successfully address your concerns and answer your questions, by providing all supporting experiments you requested and clarifying all questions you asked. The paper has also been revised accordingly.
> > >
> > > If you have any further concerns or questions, please do not hesitate to let us know, and we will respond to them timely.
> > >
> > > All the best,
> > > Authors

---

### Official Review · Reviewer_6Dgh · 2021-11-04

**Correctness:** 4
**Technical Novelty And Significance:** 3
**Empirical Novelty And Significance:** 3
**Recommendation:** 6
**Confidence:** 4

**Main Review:**

Strengths

* The paper is meticulously written, well structured, and tackles the really interesting and challenging problem learning from
 scarcely labeled data.
* The paper has good flow, it builds from the existing literature, and first points out the drawbacks of the existing methods,
 which then motivates the proposed X-model to tackle those issues.
* Extensive experiments on both classification and regression tasks show the ability of the proposed method’s ability to exploit
 strong data and model stochasticity, resulting in enhanced performance. It is shown that the proposed method is able to outperform all the methods compared in the paper.

Concerns

* One of the major concerns is regarding the technical aspect of the proposed approach. It seems to be very similar to another
 method proposed for semi-supervised domain adaptation [1], which is also proposed to improve the data efficiency of the model. While they don’t explore a task beyond classification and the problem beyond domain adaptation, the overall approach seems to be
 very similar. Furthermore, the overall idea to enhance model stochasticity has a lot of similarities with [1], [2], [3], [4] which are not mentioned in the paper at all. Hence, it is essential to have the method proposed in [1] for comparison and needs to have a discussion on the differences between the proposed method and [1]. Another experiment to consider would be Mean-Teacher + [1] which would be almost the same as the proposed idea.
* The paper does provide an analysis with the UDA method, not sure why semi-supervised DA methods are not considered as it is
 more similar to the proposed problem setup than UDA, having such comparison would strengthen the claims provided in the experimental section.

[1] Saito, Kuniaki, et al. "Semi-supervised domain adaptation via minimax entropy." Proceedings of the IEEE/CVF
 International Conference on Computer Vision. 2019.

[2] Saito, Kuniaki, et al. "Adversarial Dropout Regularization." International Conference on Learning Representations.
 2018.

[3] Saito, Kuniaki, et al. "Maximum classifier discrepancy for unsupervised domain adaptation." Proceedings
 of the IEEE conference on computer vision and pattern recognition. 2018.

[4] Lee, Seungmin, et al. "Drop to adapt: Learning discriminative features for unsupervised domain adaptation."
 Proceedings of the IEEE/CVF International Conference on Computer Vision. 2019.




**Summary Of The Paper:**


The paper focuses on reducing data labeling efforts by improving data efficiency. In contrast to most existing approaches that address this problem only in the classification setup, the paper focuses on both classification and regression set ups. The proposed method primarily is built on leveraging invariance to data stochasticity and model stochasticity. Experiments are conducted on various tasks like age estimation, key point localization, and object recognition.

**Summary Of The Review:**

My concerns are primarily due to similarities to some of the works in semi-supervised DA techniques. Considering that the paper addresses the problem in a more general setting of classification and regression, I am leaning towards accept for now.

---

> ### Author Response · Authors · 2021-11-19
> **Response to Reviewer 6Dgh**
>
>
> We appreciate the recognition of our work and the thorough review from Reviewer 6Dgh. We have clarified the questions in the following response.
>
> **Q1:** The differences between $\chi$-model with Minimax Entropy (MME) [1].
>
> Thanks for pointing out these great works, especially the most similar one: Minimax Entropy (MME) [1]. We have discussed these works in the related work and added MME as a baseline.
>
> The main differences between MME and $\chi$-model can be summarized in the follows:
> - **Setups**: As stated by Reviewer 6Dgh, MME is typically designed for classification setup, while $\chi$-model is general to both classification and regression setups.
> - **Motivations**: MME is proposed to tackle a domain adaptation problem with distributional shift, while $\chi$-model  aims at improving data efficiency in deep learning on labeled and unlabeled data with the same distribution.
> - **Technical Details**: MME uses *only one head and one data augmentation*, while $\chi$-model adopts *two heads and different data augmentations* to enhance invariance to model stochasticity and data stochasticity respectively.
> Further, MME uses the *entropy function* mainly defined in the categorical label space from the information theory, causing it difficult to be adapted into deep regression. On the contrary, the minimax loss function in $\chi$-model is general to further enhance the invariance to model stochasticity in both classification and regression setups.
>
> We further conducted experiments of MME and MME + Mean Teacher on Cifar-100 as shown in the table below.
>
> | Method  | 2500 labels | 10000 labels |
> |-----------|-----------| -----------|
> | Mean Teacher | 53.91 ± 0.57 | 35.83 ± 0.24|
> | MME | 47.40 ± 1.75 | 32.54 ± 0.81|
> | MME + Mean Teacher | 46.14 ± 1.20 | 29.64 ± 0.88 |
> | $\chi$-model (Ours) | 27.11 ± 0.65  | 20.98 ± 0.33 |
>
> Surprisingly, although MME is a state-of-the-art adversarial method mainly designed for semi-supervised domain adaptation setup, it still achieves competitive performance in our data-efficient deep learning task with a much lower error rate than Mean Teacher.
>
>
> **Q2:** Clarification of UDA in our paper with Unsupervised Domain Adaptation (UDA)
>
> We clarify that the UDA (Xie et al., 2020) [5] in our paper is a semi-supervised learning method named Unsupervised Data Augmentation (UDA) that encourages the model predictions to be consistent between an unlabeled example and an augmented unlabeled example, which is different from unsupervised domain adaptation (UDA).
>
> Actually, to improve data efficiency in deep learning, we follow the setup of semi-supervised learning by further exploring the unlabeled data in the same distribution. On the contrary, the unlabeled data in unsupervised domain adaptation comes from a related but different domain, leading to a non-independent and identically distributed (non-iid) problem.
>
> Thanks for pointing out this promising and complementary perspective to further improve data efficiency by introducing related datasets. We leave it future work to involve datasets from related domains as Unsupervised Domain Adaptation or Semi-supervised Domain Adaptation.
>
>
>
>
> [1] Saito, Kuniaki, et al. "Semi-supervised domain adaptation via minimax entropy." Proceedings of the IEEE/CVF International Conference on Computer Vision. 2019.
>
> [2] Saito, Kuniaki, et al. "Adversarial Dropout Regularization." International Conference on Learning Representations. 2018.
>
> [3] Saito, Kuniaki, et al. "Maximum classifier discrepancy for unsupervised domain adaptation." Proceedings of the IEEE conference on computer vision and pattern recognition. 2018.
>
> [4] Lee, Seungmin, et al. "Drop to adapt: Learning discriminative features for unsupervised domain adaptation." Proceedings of the IEEE/CVF International Conference on Computer Vision. 2019.
>
> [5] Qizhe Xie, Zihang Dai, Eduard Hovy, Minh-Thang Luong, and Quoc V Le. Unsupervised data augmentation for consistency training. NeurIPS, 2020.

---

> > ### Author Response · Authors · 2021-11-30
> > **Request feedback from Reviewer 6Dgh**
> >
> > Dear Reviewer 6Dgh,
> >
> > Many thanks for your time and efforts in reviewing our paper.
> >
> > We have made an extensive effort to try to successfully address your concerns by providing all supporting experiments you requested and clarifying all questions you asked. The paper has also been revised accordingly.
> >
> > It will be helpful if you can provide some feedback for us to know whether our responses have addressed your concerns.
> >
> > If you have any further concerns or questions, please do not hesitate to let us know, and we will respond to them timely.
> >
> > All the best, Authors

---

### Author Response · Authors · 2021-11-19
**Summary of Revisions**

We appreciate all three reviewers for their insightful and constructive comments.
We have uploaded a revised draft to address all reviewers' comments. The main changes (noted by red color in the paper) are summarized below:

- We have modified the Related Work part to discuss with some classical and various recently proposed methods mentioned by the reviewers, including Meta Pseudo Labels, MixMatch, UDA, ReMixMatch, Co-Training, Deep Co-Training, Tri-Training to make the survey of SSL literature more comprehensive.

- In the Experiment part, we have added more experimental results based on the reviewers’ comments, including Meta Pseudo Labels, Co-Training, Deep Co-Training, and MC-dropout, while MixMatch, UDA, ReMixMatch have been added as baselines in the first draft.

We hope our responses and revisions will address all reviewers’ concerns!

---

### Decision · Program_Chairs · 2022-01-20

**Decision:**

Accept (Poster)

**Comment:**

The SAC wrote a very good meta review and I just copy and paste it here. I completely agree with the SAC that the contribution of the paper due to the similarity to MME and MCD. Hopefully adding data augmentation to MCD and providing empirical results on new tasks can shed some lights to the community.

--------------------
Based on a request from the authors, the SAC read the paper and the reviews and engaged two additional expert reviewers to provide an additional assessment.

The paper addresses semi-supervised learning (SSL) and makes two contributions: 1) a method called χ-model, which combines data augmentation with a two-headed network that has an adversarial loss between the heads and the feature backbone; 2) an empirical evaluation on classification and regression SSL tasks.

As pointed out by reviewer 6Dgh, the proposed model is very similar to two existing methods: MME and MCD. These related works were not mentioned in the submission. During the rebuttal the authors compared to MME but not MCD.

Similarity to MME: The core idea in MME (Min-Max Entropy) is to introduce a min-max game between the head and backbone to regularize the model. The authors point out that one difference is that X-model uses two heads and regression loss, while MME uses a single head and entropy loss. They also have other components like data consistency regularization, which are borrowed from prior work (FixMatch). They further point out that MME was evaluated and motivated for unsupervised domain adaptation whereas their paper evaluates on the same distribution (regular SSL). In general, there is a lot of overlap and borrowed techniques between UDA and SSL methods, but the experimental benchmarks tend to be different.

Similarity to MCD: A larger concern is that the proposed method is actually more similar to MCD than MME. In Eq.5, the data is augmented into two ways and two different output heads are applied. If the data is not augmented this way, the formulation is the same as MCD. The authors did not discuss this point, even though reviewers pointed out the similarity. From the perspective of the technical approach, the novelty wrt MCD appears to be mainly in adding the data augmentation. They do show in the paper that this type of min-max regularization is effective in various regression tasks, which is a good empirical contribution. In the ablation studies of the data augmentation part, the difference over MCD is not very large, but the augmentation provides good gains in some experiments (although MCD is not mentioned, in Table 2 and 5, the "χ-model (w/o data aug.)" is likely essentially MCD.)

Overall, the strength of the paper appears to be in adding data augmentation to MCD and providing empirical results on new tasks showing that it works on same-distribution test sets and on regression tasks. The technical contribution seems somewhat limited, if accepted, the paper should include a clear discussion w.r.t MCD in the method and experiment sections. Furthermore, some baselines may potentially be missing, e.g. an MDD-based (similar to MCD) UDA method for regression can be used as a baseline (Regressive Domain Adaptation for Unsupervised Keypoint Detection, CVPR'21).